# A Virtual Instrument for Road Vehicle Classification Based on Piezoelectric Transducers

**DOI:** 10.3390/s20164597

**Published:** 2020-08-16

**Authors:** Bernardino González, Francisco J. Jiménez, José De Frutos

**Affiliations:** 1Escuela Técnica Superior de Sistemas en Ingeniería de Telecomunicación (ETSIST), Universidad Politécnica de Madrid, Carretera de Valencia, km 7, 28031 Madrid, Spain; 2Departamento de Electrónica—Física, Ingeniería Eléctrica y Física Aplicada, Universidad Politécnica de Madrid, Carretera de Valencia, km 7, 28031 Madrid, Spain; franciscojavier.jimenez@upm.es (F.J.J.); jose.defrutos@upm.es (J.D.F.); 3Cemdatic-Poemma R&D Group, Avenida. Complutense, 30, 28040 Madrid, Spain

**Keywords:** virtual instrumentation, vehicle recognition, wheelbase, LabVIEW^®^, piezoelectric transducers

## Abstract

This paper describes a virtual instrument capable of the automatic and quasi-instantaneous classification of a vehicle according to category when it is driving along the road. The vehicle’s classification is based on accurate measurements of both the vehicle’s speed and its wheelbase. Our research is focused on achieving accurate speed and wheelbase measurements and then determining the category of the vehicle through the developed software. The vehicle categorization is based on the wheelbase measurements and the number of axles and metal masses of the vehicle. The system has a complementary magnetic sensor, which helps in classifying the vehicle when the wheelbase measurement could be representative of different categories, and a *camera* to confirm the results of the experiment. The proposed measurement system presents a novel method for classifying vehicles according to type using piezoelectric transducers (piezo sensors). In addition, no system references have been found that encompass the functionalities of the presented system based on the information of only two piezoelectric transducers. The system has important advantages over current alternatives (systems based on inductive loops, cameras, fiber optic sensors or lasers), the installation is simple and non-invasive and with a success rate of the classification greater than 90%. The system consists of a signal acquisition point on the pavement, signal conditioning hardware and a data acquisition (DAQ) module, which links the hardware and the virtual instrument developed in LabVIEW^®^. Finally, the system has been tested on the road with real traffic, and the experimental results are presented and discussed in this paper.

## 1. Introduction

In recent decades there has been a considerable increase in intelligent electronic systems applied to vehicles and infrastructure. The evolution of Intelligent Transportation Systems (ITS) has introduced a wide range of systems and software applications [1] with the aim of improving safety and efficiency in road transport. These systems obtain information from the different elements of the roads and process it and are used to improve driver safety, traffic and travel comfort.

Among others, ITSs are present in the development of:Systems to increase driver safety, such as Advanced Driver Assistance Systems (ADAS) [2,3,4].Traffic information systems, variable signage panels, Vehicle-to-Infrastructure (V2I) communication or free-flow tolls.Systems to improve traffic efficiency, vehicle classification and counting systems [5,6,7,8,9]. Automatic vehicle classification systems are important for public administrations and private companies that demand systems that carry out a classification of vehicles for dynamic charging for infrastructure use to promote sustainable mobility.

In so-called “smart cities” and “smart roads” [10], it is currently of great interest to know the quantities and categories of vehicles that are circulating in “real time”. For this reason, within the scientific technology of electronic instrumentation and measurement and control systems, there are different lines of work [1,2,3,4,5,6] focused on meeting this demand.

This line of research pursues synergy between the data provided by the installed instrumentation equipment and the existing infrastructure to achieve more sustainable and environmentally friendly mobility. The ITSs in large cities obtain information from the vehicles that circulate in them, and this is where virtual instrumentation plays an essential role.

Although many types of sensors are used to meet this technological need, the most widely used on the market today are magnetic (inductive loops or inductive sensors) or based on the digital processing of images captured by cameras or fiber optic sensors. Cameras, magnetics and inductive systems are the most used in the city of Madrid, where the *virtual instrument* described and tested in this paper was developed.

In the case of inductive sensors, the main problems are, firstly, the uncertainty in the results of the classification. These devices do not allow the classification of multi-axle vehicles by themselves and require other elements to count them, such as photocells, fiber optic sensors, laser sensors or axle-counting stands. The inductive sensors are based on the variation of the mutual inductance (L) during the passage of the vehicle [9]. The mutual inductance L can vary depending on whether the vehicle is loaded with additional metal mass. The second problem is that the installation of sensors is usually intrusive, i.e., when the sensors are placed under the pavement so that the vehicles pass over them. This makes the cost of installation high.

Classification systems based on image processing have some basic problems: The first is the need to place the camera on an elevated support, such as a pole or a gantry. The second problem is the large volume of data to be processed, which makes the detection and classification slow. Another problem is that the location of the camera directly influences the calculation parameters in the detection algorithm [11], and this makes the settings required in each installation variable. In short, the installation of this type of system and the processing of information are complex.

Fiber optic-based systems [12] analyze the vibration produced by the vehicle and require complex signal processing and analysis algorithms. The success rate is less than 80%.

Consequently, the motivation of the authors is to create an alternative system that can minimize the problems described.

The objectives established when designing and building the virtual instrument described in this paper can be summarized as follows:Sensors should be easy to install, preferably with non-invasive installation.When sensors are to be installed under the pavement, it should be possible to do so with little impact on the road (in terms of road works or service interruption).Data processing should be fast, with low computational costs.It should be possible to extend the functionalities to the system without the need to increase the number or type of sensors.Measurements should be sufficiently accurate to reduce the probability of error in the vehicle category classification.Signal conditioning and data acquisition hardware should have low costs and low power consumption.The system must be able to catalog multi-axle vehicles.The system must store the collected data for later use in a back-office process.

The description of the designed and built system has been organized as follows: System blocks: The system blocks are described, and their nature, hardware or software is specified. The virtual instrument and its blocks are defined. The installation of the measuring system designed for the road is described.System hardware: All hardware elements of the measurement system are described: transducers, signal conditioners, data acquisition (DAQ) systems (DAQ module) and camera.System software: The software architecture (pattern design) of the developed application is described. The application developed and the calculation methodology used to achieve the results are described.Experimental results: The results obtained are shown and the results are discussed.Error estimation: An analysis of the sources of error is carried out and the estimate of the maximum error is calculated.Conclusions: The final conclusions of the designed system are shown after being tested with real traffic.

## 2. Built Measurement System Blocks

The system was conceived as a virtual instrument consisting of a hardware part and a software part running on a computerized system (PC), so that changes in the software can make the virtual instrument expandable in its functionalities and maintainable.

A virtual instrument consists of an industry-standard computer or workstation equipped with powerful application software, cost-effective hardware such as plug-in boards, and driver software, which together perform the functions of a measurement instrument [13].

Virtual instruments represent a fundamental shift from traditional hardware-centered instrumentation systems to software-centered systems that exploit the computing power, productivity, display and connectivity capabilities of popular desktop computers and workstations.

In the virtual instrumentation, the “software is the instrument” [14]. The software, using the appropriated hardware, defines the functions of the instrument.

In the Figure 1, the virtual instrument block is formed by the developed software and a National Instruments general purpose multi-function DAQ module. Obviously, signal conditioners for the transducers are also necessary. The system block diagram is shown in Figure 1.

Figure 2 shows an example of installation. Piezo sensors can be installed above the pavement (non-intrusive) or below the pavement (intrusive).

## 3. System Hardware

The following describes the transducers used, their signal conditioners, the data acquisition (DAQ) board used and the camera used to capture the images.

### 3.1. Transducers

#### 3.1.1. Piezo Sensors

The piezoelectric transducers used are piezoelectric cables. These piezoelectric devices are commonly used in traffic management and have, in recent years, also been used in the development of road traffic infrastructure [15,16,17,18,19,20,21]. This type of transducer is usually used in electronic instrumentation dedicated to traffic management for vehicle counting and weighing (only if the vehicle is traveling at low speed). However, they are used in a new way in this design to measure the speed of the vehicle with high accuracy. The procedure for obtaining the speed and wheelbase measurements is described in Section 5.

Piezoelectric cables [22,23] are coaxial with 20 AWG stranding in which the cable insulation is a PVDF film. The most important characteristics are shown in Figure 3 and Table 1. Piezoelectric performance specifications.

Piezo sensors, due to the direct piezoelectric effect of the PVDF piezo film tape, respond with an analog voltage. The peak values when the vehicle passes over them depend on whether the cable is on or under the pavement. The margins are 20~40 V when the installation is on the pavement and 2~4 V when the installation is under the pavement, at a 2 cm distance. It is necessary to attenuate and adapt this value so that the signal is within the input range of the DAQ module. Figure 4 shows a typical response signal from the piezo sensors when they are driven over by the vehicle.

#### 3.1.2. Magnetic Sensor

A magnetoresistive transducer [24,25], the HMC2003 transducer of the HMC series of Honeywell, is used. It is able to determine the variations in the magnetic field of the Earth induced by the approach and presence of metallic masses in the three axles—X, Y, Z. Figure 5 shows a block diagram of the magnetic sensor, and some of the specifications are shown in Table 2.

These three axles use permalloy magnetoresistive sensors. These sensors produce an imbalance in the Wheatstone bridge (see Figure 6. HMC2003 set–reset.) when the magnetic dipoles are oriented to the passage of the metal mass. Two control actions are required to achieve the maximum sensitivity and input range from these sensors. The first is the establishment of a set–reset command (introducing SR+ and SR− pulses) [24,25] to disorientate the magnetic dipoles and thus “erase” the memory of the last measurement (the dipoles tend to remain oriented). The second is the correction of the offset voltage to approximately 2.5 V when there is no metal mass to unbalance the Wheatstone bridge. The control actions are dictated by the software through the DAQ module. Digital set–reset control signals are applied to a driver made up of MOSFET transistors that introduce two SR+ and SR− pulses. An analog voltage is generated, which is subtracted from the output voltage in the absence of metal mass. Figure 7 shows the signals of a vehicle of three axles: magnetic sensor, piezo sensor 1 and piezo sensor 2 after its conditioning. The shape and area under the magnetic sensor graph provide additional information for distinguishing between doubtful cases in the category classification, although this procedure is still being developed.

#### 3.1.3. DAQ Module

A National Instruments general purpose multi-function DAQ module is used [26], NI USB 6211 [27]. Some of its features are shown in Table 3 and Figure 8.

#### 3.1.4. Camera

The system has a camera that takes an image of the vehicle. This image is associated with the measurements taken. The initial objective is to verify that the classification method implemented in the IV works correctly. A Logitech C920 HD 1080P USB camera connected to the PC is used.

Figure 9 shows images of the built equipment.

The sensor signal conditioners (for piezo sensors and magnetic sensor) were designed by the authors. For each piezo sensor, an active attenuator was designed and for the magnetic sensor, an amplifier, MOSFET drivers to set–reset the signal and a circuit to adjust the zero level in the magnetic sensor response were designed.

## 4. Software System

The system’s software was written with LabVIEW^®^ (National Instruments, Austin (Texas) USA), a graphical programming language that is the most widely used language in virtual instrumentation. The LabVIEW^®^ program consists of two main windows, the front panel (user interface) and the block diagram (program code). The programming is modular, and each module is built with one or more files called “SubVIs” [28].

The virtual instrument for measurement and control was designed with a state machine architecture. This design pattern is indicated for the design of complex measurement equipment that is required to have different functionalities. A state is a situation in which the machine can be found. Usually, in a state, the actions or operations associated with it are carried out. There is also a decision code, which determines the next state to which the machine must evolve. A state machine is described graphically with a state diagram in [29]. The LabVIEW^®^ programming language allows for the implementation of a state machine with a simple design pattern, which allows for the development of complex programs, making the virtual instrument scalable, maintainable and readable.

Figure 10 shows the skeleton of a state machine where the strategy described in the previous paragraph can be clearly seen.

Figure 11 shows the state diagram of the built virtual instrument.

The states of the virtual instrument are described below and shown in Table 4. Virtual instrument states.

### 4.1. Speed Vehicle Calculation

The designed virtual instrument has two keys that ensure a reliable measurement with a high degree of accuracy:-Acquisition of the signal from the piezoelectric sensors at an acceptable level.-Highly accurate calculation of vehicle speed.

Speed, *v*, is calculated using Equation (1):(1)$$v=\frac{d}{Delay_{2,1}}\times 3.6$$
where
v is the speed, expressed in km/h;d is the distance between the piezoelectric sensors, expressed in meters (m);Delay_2,1_ s the time from the vehicle passing over piezo sensor 1 to the vehicle passing over piezo sensor 2, expressed in seconds (s).

Figure 12 shows the signals captured by piezoelectric sensors 1 and 2, respectively. As can be seen, the signals are similar but vary in shape and amplitude. These factors make the uncertainty in the estimation of the Delay_2,1_ quite high (in the order of ms). Therefore, the need arises to perform the Delay_2,1_ estimation by a method that provides a lower error [30].

The estimation method used is cross correlation, which considerably reduces the error rates in the estimation of the delay between signals [31,32]. This has been implemented in SubVI Cross correlation.vi.

Figure 13 shows the code extract from Cross correlation.vi. This SubVI has input data (signals) acquired from piezoelectric sensors 1 and 2 and the distance between them. The SubVI provides a graph of the cross correlation and the estimated speed (in km/h).

Although the signals from the piezo sensors were conditioned beforehand, the sensitivity of each piezo cable does not match; in the same way, the acquired signal levels also do not match. The solution implemented to make the acquired signals from the sensors valid is the standardization of these signals.

The next step is to determine the cross correlation of both signals, which is calculated according to Equation (2):(2)$$R_{\mathrm{xy}}\left(t\right)=x\left(t\right)⊗y\left(t\right)=\int_{-\infty }^{+\infty }x^{*}\left(\tau \right)\cdot y\left(t+\tau \right)d\tau$$
where x(t) is signal piezo 1 (normalized) and y(t) is signal piezo 2 (normalized). However, since the signals are discrete, Equation (3) is implemented.
$$x_{j}=0,j<0\;\mathrm{or}\;j\geq N\;\mathrm{Signal}\;\mathrm{Piezo}\;1\left(\mathrm{sampled}\;\mathrm{and}\;\mathrm{normalized}\right)\;y_{j}=0,j<0\;\mathrm{or}\;j\geq M\;\mathrm{Signal}\;\mathrm{Piezo}\;2\left(\mathrm{sampled}\;\mathrm{and}\;\mathrm{normalized}\right)$$
(3)$$h_{j}\left(t\right)=\overset{N-1}{\underset{k=0}{\sum }}x_{k}^{*}\cdot y_{j+k}\;\mathrm{for}j=-\left(N-1\right),-\left(N-2\right),\ldots ,-1,0,1,\ldots ,M-2),\left(M-1\right)$$

The implementation of this equation provides the sequence of expression (4):(4)$$R_{{\mathrm{xy}}_{i}}=h_{i-\left(N-1\right)}\mathrm{for}i=0,1,2,\ldots ,N+M-2.$$

The result of cross correlation for signal piezo sensor 1 and signal piezo sensor 2 is shown in Figure 14. Therefore, the delay between the vehicle steps according to the piezoelectric sensors is calculated using Equation (5).
(5)$${\mathrm{Delay}}_{2,1}=t_{2}-t_{1}=t_{\mathrm{corr}}-t_{\max}$$
where
t_1_ and t_2_ are the moments of time at which the vehicle moves over piezoelectric sensors 1 and 2, respectively.t_corr_ is the waveform duration of the cross correlation result;t_max_ is the moment in time at which the cross correlation reaches the maximum value.

### 4.2. Vehicle Category Determination

The automatic classification of the vehicle category is done by estimating the number of axles of the vehicle and the wheelbase or the distance between axles. The category is then assigned as shown in Table 5. This strategy is novel compared to other methods for the classification of vehicles used by other authors [33].

The classification shown in Table 5 is based on a classification made according to the size of the vehicle and published in Traffic Detector Handbook Vol. 1 [7].

The wheelbase categorization strategy is one of the key factors for achieving a high probability of success regarding the system’s automatic classification.

The calculation of the distance between two axles of the same vehicle is done via Equation (6):(6)$$d_{\mathrm{axles}}=\frac{v}{{\mathrm{Delay}}_{i,i+1}}$$
where
d_axles_ is the distance between two consecutive axles, expressed in meters (m);v is the speed, expressed in km/h;Delay_i,i+1_ is the time from when the i axle of the vehicle drives over a piezoelectric sensor until the i + 1 (next) axle drives over the same sensor, expressed in seconds (s);Delay_i,i+1_ is estimated by measuring the time instants of the signal from one of the sensors when a certain threshold value is exceeded. In this case, Delay_i,i+1_ is not calculated using the cross correlation method. This would necessitate adding another sensor to establish the signal start trigger. The classification of the vehicle is done based on ranges of the vehicle wheelbase, as shown in Table 3. These calculations are performed in the SubVI Axles distance.vi.

Figure 15 shows a code extract from Axles distance.vi. This SubVI has, as interesting input data, the signals acquired from the piezoelectric sensors 1 and 2, the selection of the observed sensor (piezo sensors 1 or 2) and the estimated speed in (km/h) calculated by the SubVI Cross correlation.vi, described in the previous section. The data of interest provided by Axles distance.vi are the value of the number of axles of the vehicle (#founds), an array with the instants of the time of step (locations) and the array with the distance between the axles of the vehicle (distance axles array (m)).

The data for the distance axles array (m) and #founds are used to determine the vehicle category.

Regarding the vehicle category, a study was performed of the wheelbase of common vehicles circulating in Spain. The authors added the typical wheelbase and the result is the information shown in Table 5.

The subVI in charge of performing the automatic classification is called Category.vi. Figure 16 shows the flowchart of this subVI.

When Category.vi returns the value unknown, a sensor measurement error exists, or the measured values are not included in the values of Table 5. Remember that the values in Table 5 are written in the file Categories.ini (ASCII file). In any case, the measured values of the number of axles (#founds), the distance between axles (distance axles array (m)) and the photograph of the vehicle are always recorded. This allows the user to check, at a later stage, if there are errors in the sensors or if the sensor measurements recorded a vehicle not covered by the categories in Table 5. If you want to add new vehicle categories, just edit the file Categories.ini.

Figure 17 shows a vehicle with a wheelbase of approximately 2.57 m, corresponding to the urban/subcompact category.

## 5. Experimental Results

### 5.1. Test Conditions on Road

Tests were conducted with actual road traffic according to the recommendations given in ASTM 1318-09 regarding geometric design and pavement conditions [34]. To ensure the alignment of the different elements on the road, two laser levels and their signal receivers were used, ensuring an error of ± 0.01 m in the measurement of the distance between piezo sensors 1 and 2.

The test was carried out as shown in Figure 18. 

The piezo sensors were mounted non-intrusively (on the floor), and the piezo cable was covered with a ProTech-branded cable protector for the floor. Below is an example (Figure 19) of a vehicle measurement, resulting in a “rigid truck (three axles)” classification.

### 5.2. Proccessing/Compute Time

LabVIEW^®^ profile performance [35] was used to quantify the software processing time when performing the measurements.

In the measurement process, the execution times of the subVIS involved in the machine states were quantified when carrying out the measurement (see Table 6). Ten runs were done to estimate the processing time.

Stand by status is a waiting status and does not affect the processing time to present the measurement.

Configure DAQ status is a state prior to reading the sensors and therefore does not affect processing.

Set–reset status is executed every time a measurement is made and has a fixed execution time of 25 ms, this is the time programmed for the digital pulse applied to the magnetic sensor.

The time between average measurements can be estimated as T_between measurements_ (ms) = 655 + 7.9 + 25 = 687.9

## 6. Error Estimation

This section concerns the estimation of the error in the measurements, speed and distances between the axles provided by the virtual instrument.

The sources of error were estimated based on the characteristics of the equipment used and tests carried out in the laboratory.

In terms of the speed measurement, three significant sources of error were determined:
Inaccuracy in the measurements between wires, d_b_.The different waveforms in the received signals, which can cause a maximum shift in the calculation of the cross correlation.The error due to the signal sampling instant that can cause the maximum point of the cross correlation to be shifted by Ts = 1/f_s_, where f_s_ is the sampling frequency for the acquisition of the analog signals from piezo sensor 1 and piezo sensor 2.

Regarding the wheelbase measurement, two significant sources of error were determined:The uncertainty in the speed measurement.The uncertainty in the Delay_i,i+1_ calculation (defined in Section 4.1), which depends on how accurately the moment at which the vehicle passed over the same piezo sensor for the axle (axle_i_) and the adjacent axle (axle_i+1_) was detected. This calculation error is low since the slew rate of the signals in the piezo sensors is approximately 1.5 V/ms.

Table 7 and Table 8 show the estimated errors for a two-axle vehicle with wheelbase 1 = 1.375 m and wheelbase 2 = 4375 m, respectively, for speeds between 25 km/h and 175 km/h. Loose error values are estimated for the above error sources, and these are as follows:Distance between piezo sensors, d_b_ = 6 m. Error_db_ = ±0.01 m.Error due to signal waveform, Error_wave_ = ±100 µs.Error due to instant sampling, f_s_ = 20 kHz. Two-sample error: Error_fs_ = ±100 µs.The uncertainty in the calculation of Delay_i,i + 1_, Error_Delayi,i + 1_ = ±1 ms.

In order to estimate the errors due to unrecognized categories or wrong measurements (“Unknown” in Figure 16. Category.vi.) the data from the road tests were analyzed. Two tests were carried out at road M-404 milestone 1 in Madrid (Spain) and two tests at road N-232 milestone 492 in Castilla-León. A total of 457 measurements were taken and the results are shown in Table 9. 

“Unknown measurement NO OK” is the set of vehicles that drove over only one of the piezo sensors (piezo sensor 1 or piezo sensor 2) or there was a wrong connection at the beginning of the test that was later resolved. Only three vehicles were detected that were not assigned to any category in Table 5.

## 7. Conclusions

The designed and built virtual instrument is capable of performing the quasi-instantaneous classification of vehicles circulating on real traffic roads.

The installation can be performed in an intrusive way with a simple intervention on the pavement. A non-intrusive installation can also be carried out, as described in Section 5.

The system is capable of taking measurements with a low level of error in speed measurements and an acceptable level of error in wheelbase measurements.

A classification strategy based on the vehicle wheelbase and number of axles was implemented, allowing for the rapid classification of the most common vehicle types.

A magnetic sensor was included, which generates a magnetic “footprint” that provides information about the type of vehicle. We are developing a line of research regarding this footprint, which complements the classification of the vehicles by category. The magnetic “footprint” can be used to elucidate doubtful cases that may arise in the analysis of the data measured with piezo sensors.

The built hardware and the commercial hardware used are basic performance equipment, so their prices are not high.

The design of the virtual instrument has a defined software architecture. A state machine was used and care was taken in the programming style so that the software is readable, expandable and maintainable. In fact, it would be easy to expand the software to replicate the system in all lanes of a road.

The virtual instrument’s user interface is user friendly and simple, making it easy for the system operator to use. The test results are exportable since the data are also saved in text format (ASCII).

## Figures and Tables

**Figure 1 sensors-20-04597-f001:**
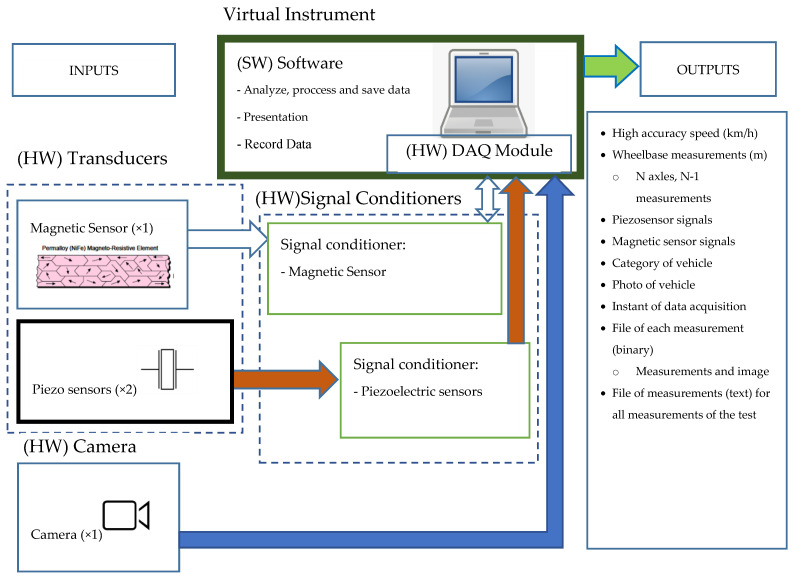
Block diagram of the measurement system. Software (SW): Virtual instrument software. Hardware (HW): Transducer, Data Acquisition (DAQ) module, signal conditioners, camera.

**Figure 2 sensors-20-04597-f002:**
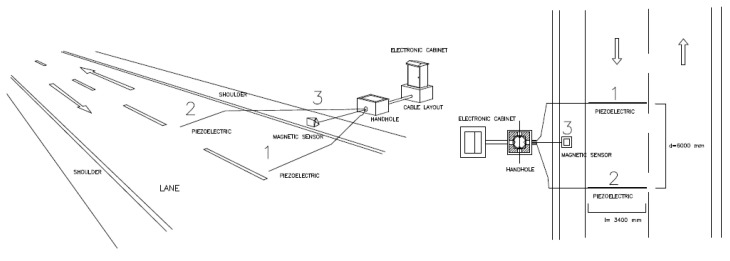
Installation model for the measuring system.

**Figure 3 sensors-20-04597-f003:**
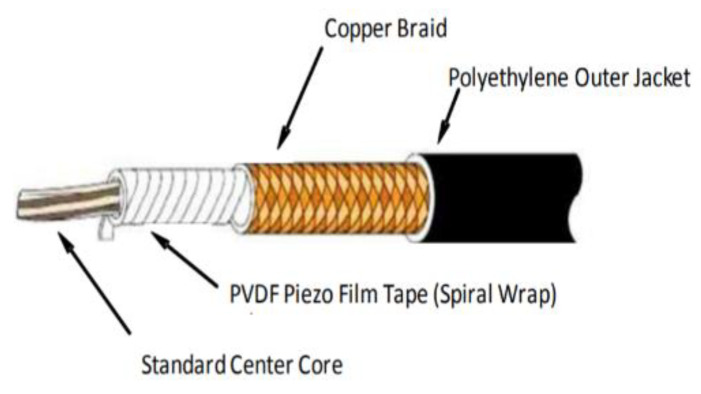
20 AWG Cable—Spiral wrap.

**Figure 4 sensors-20-04597-f004:**
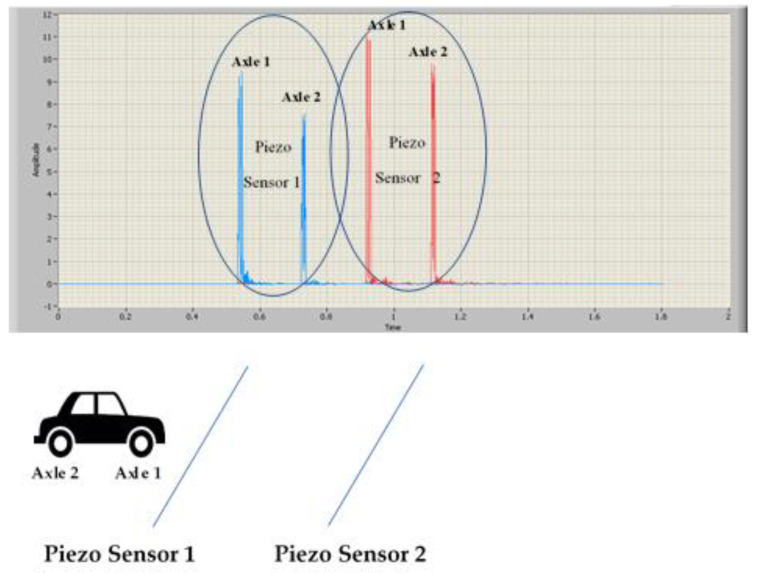
Sensor response signal (V) when sensors are driven over by a vehicle.

**Figure 5 sensors-20-04597-f005:**
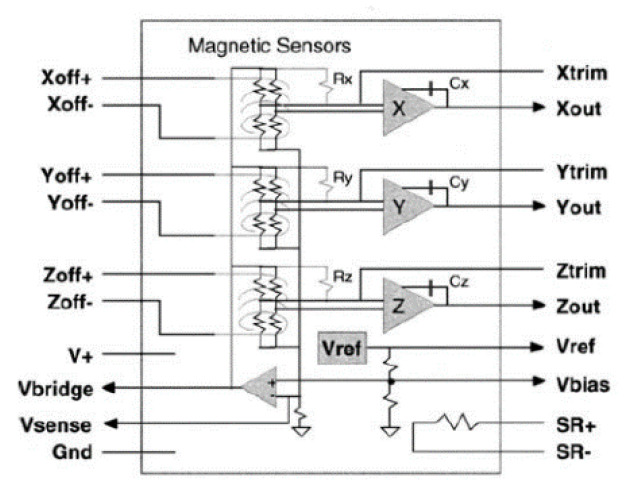
HMC2003 block diagram.

**Figure 6 sensors-20-04597-f006:**
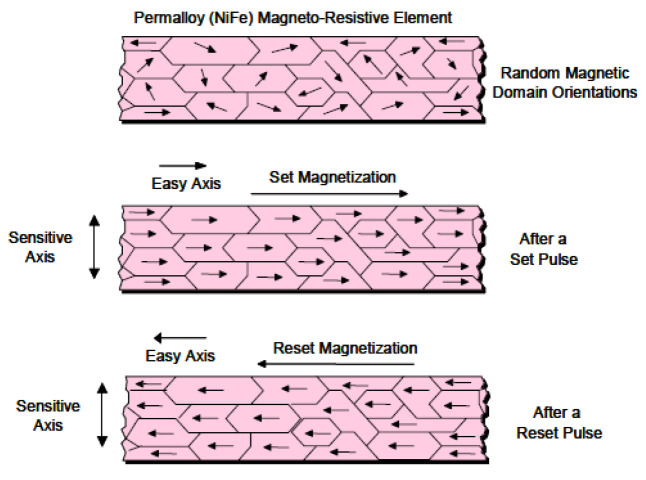
HMC2003 set–reset.

**Figure 7 sensors-20-04597-f007:**
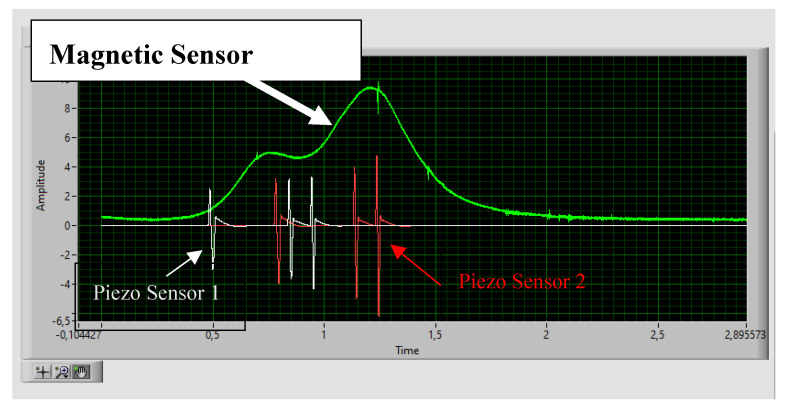
Magnetic sensor signal (V) and piezo sensors (V) in a three axles vehicle.

**Figure 8 sensors-20-04597-f008:**
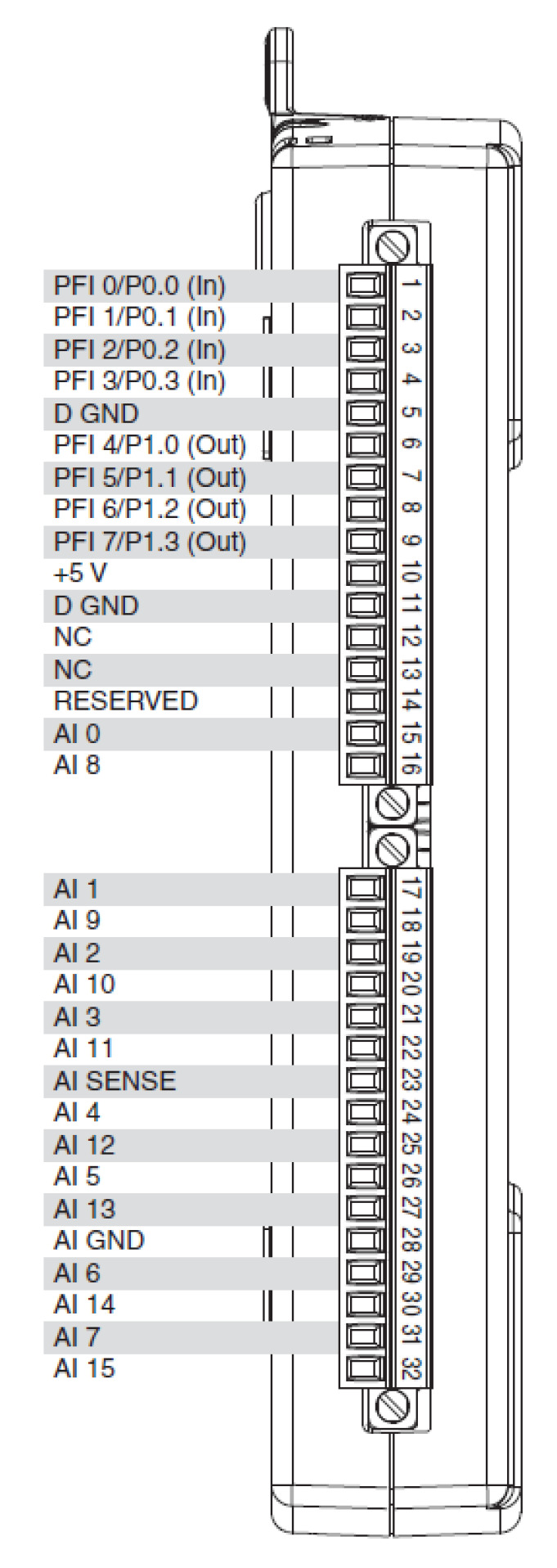
Data acquisition (DAQ) module *National Instruments USB 6211 specifications*.

**Figure 9 sensors-20-04597-f009:**
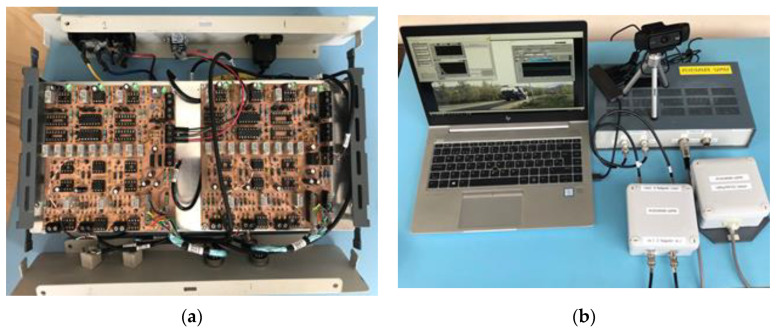
Hardware system: (**a**) sensor signal conditioning and Data Acquisition (DAQ) modules; (**b**) piezo sensor attenuators, magnetic sensor connections, camera and PC.

**Figure 10 sensors-20-04597-f010:**
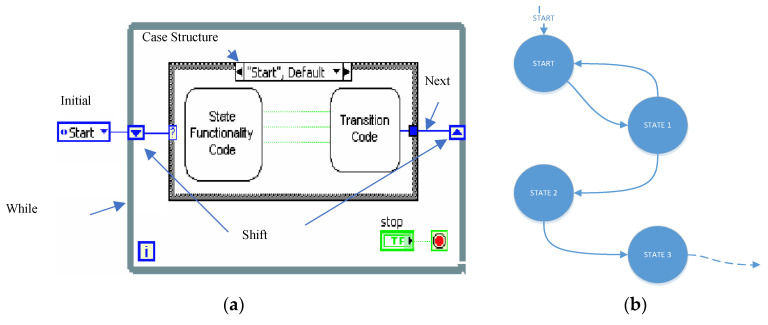
(**a**) State machine LabVIEW pattern. (**b**) Typical example of a state diagram.

**Figure 11 sensors-20-04597-f011:**
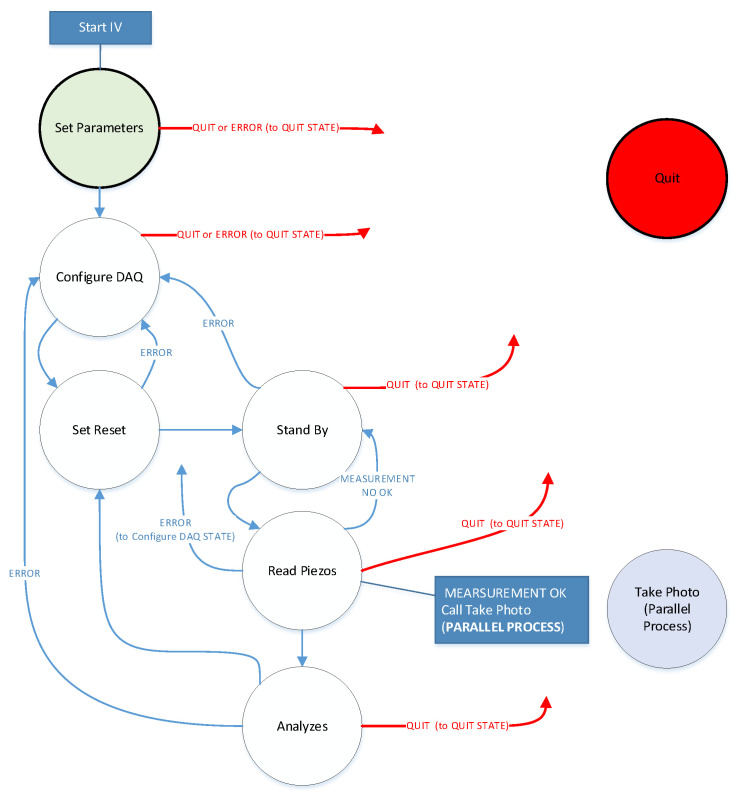
State diagram of virtual instrument.

**Figure 12 sensors-20-04597-f012:**
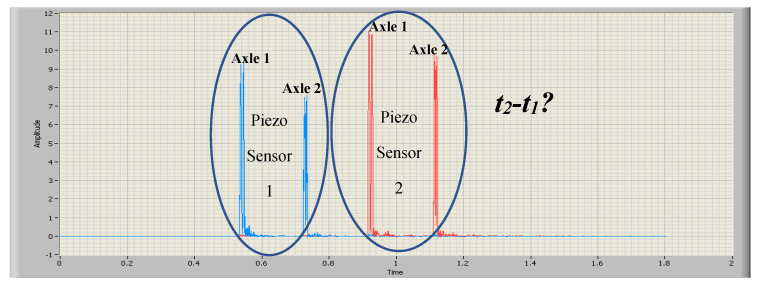
Signal of piezo sensor 1 and signal of piezo sensor 2. Two-axis vehicle.

**Figure 13 sensors-20-04597-f013:**
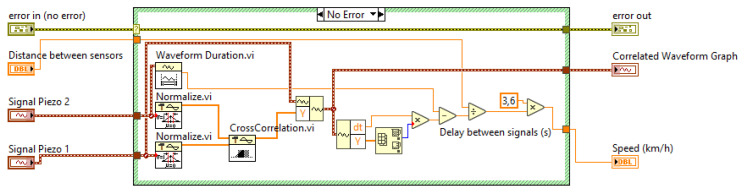
High-accuracy speed measurement, *Cross correlation.vi*.

**Figure 14 sensors-20-04597-f014:**
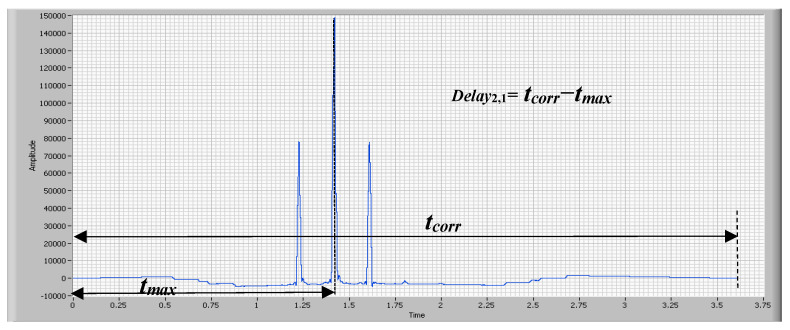
Cross correlation for signal piezo sensor 1 and signal piezo sensor 2.

**Figure 15 sensors-20-04597-f015:**
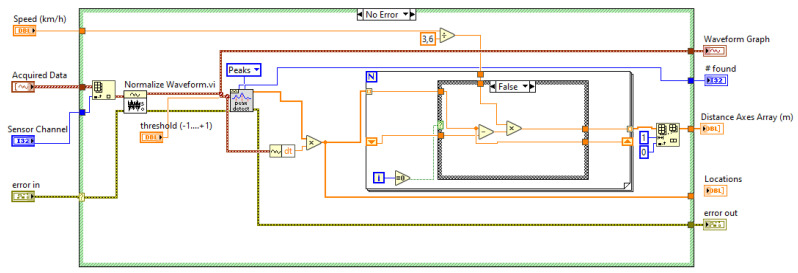
Axles distance.vi.

**Figure 16 sensors-20-04597-f016:**
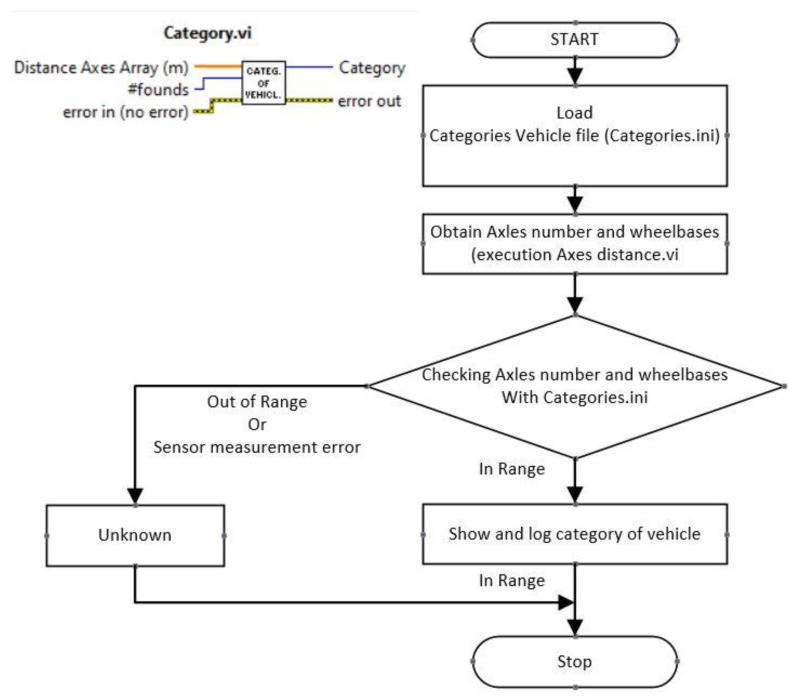
Category.vi.

**Figure 17 sensors-20-04597-f017:**
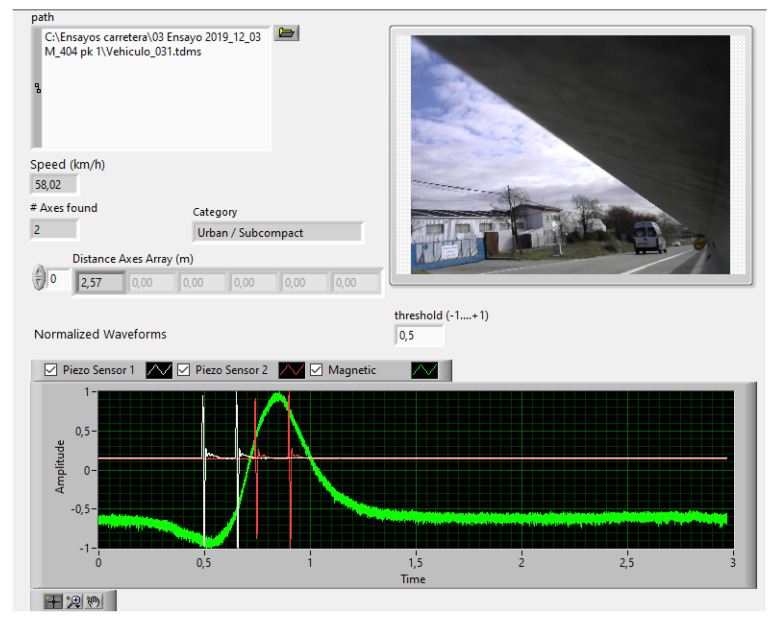
Urban/subcompact. Front panel of the virtual instrument. Amplitude signals normalized to ±1.

**Figure 18 sensors-20-04597-f018:**
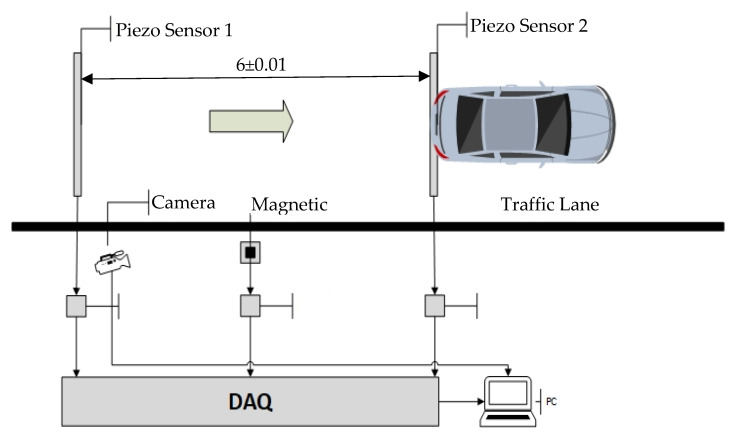
Testing system installed on road.

**Figure 19 sensors-20-04597-f019:**
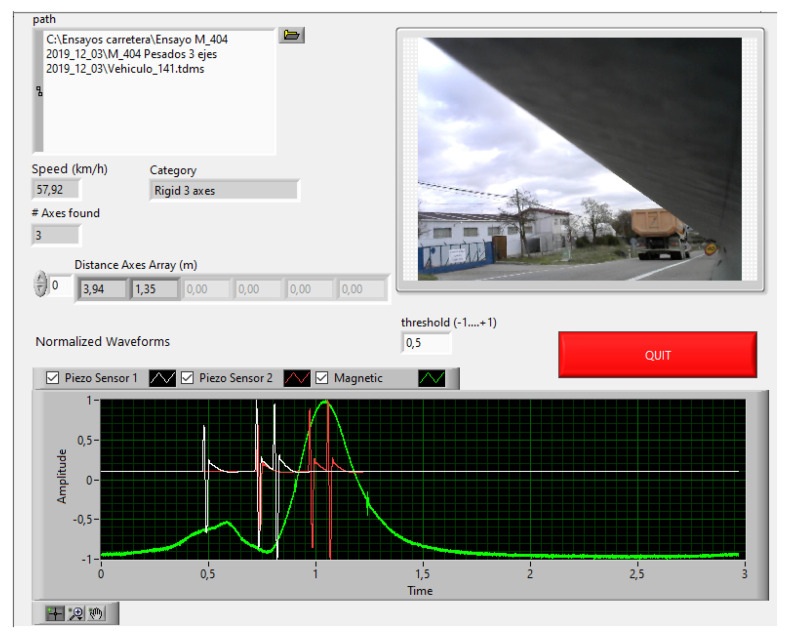
Three rigid axles. Front panel of virtual instrument. Amplitude signals normalized to ± 1.

**Table 1 sensors-20-04597-t001:** Piezoelectric performance specifications.

Properties	Value	Units
Outside Diameter	2.69	mm
Center Core	1.02	mm
Capacitance @ 1 kHz	950	pF/m
Resistance (shield)	47	Ω/km
Hydrostatic Piezo Coefficient	20	pC/N
Resistance (Center Core)	31	Ω/km

**Table 2 sensors-20-04597-t002:** HMC2003 performance specifications.

Properties	Min	Typ	Max	Units
Sensitivity	0.98			V/Gauss
Null Field Output	2.3	2.5	2.7	V
Resolution	40			μGauss
Field Range	−2		2	Gauss
Output Voltage	0.5		4.5	V
Bandwidth	1			kHz

**Table 3 sensors-20-04597-t003:** National Instruments USB 6211 specifications.

I/O	Value	Units
Analog Inputs	Eight differential or 16 single-ended	-
Analog Input Resolution	16	bits
Input Range	±10	V
Maximum Input Rate	250	KS/s
Analog Outputs	2	
Maximum Output Rate	250	KS/s
Maximum Output Range Rate	10	V
Digital Inputs	4	
Digital Outputs	4	

**Table 4 sensors-20-04597-t004:** Virtual instrument states.

Name	Description/Function	Next State
Set Parameters	Set configuration and test parameters	No Error: Configure DAQ Error: Quit
Configure DAQ	Configure the hardware system	No Error: Set–Reset Error: Quit
Set–Reset	Set a pulse signal to magnetize the magnetic sensor	No Error: Stand by Error: Configure DAQ
Stand by	Wait for vehicle detection	No Error: Read Barriers (if vehicle detected) Error: Configure DAQ
Read Piezo	Measure the piezo sensors These signals will be used in the next state to calculate the vehicle speed Give order to take photo	No Error: Analysis Error: Configure DAQ
Analyze	Calculate the vehicle speed (high accuracy) Calculate the axis distance and vehicle category Present and save data measurement	No Error: Set–Reset Error: Configure DAQ
Take Photo	Parallel process execution so as not to interfere with the rest of the actions in the IV	Waiting for next order to take photo
Quit	Release resources in an orderly manner and finish the IV There is a quit “button” on the front panel of the IV	No Error: Read Barriers (if vehicle detected) Error: Configure DAQ

**Table 5 sensors-20-04597-t005:** Axle distance category table (wheelbase).

Vehicle Type	Wheelbase 1 (mm)	Wheelbase 2 (mm)	Wheelbase 3 (mm)	Wheelbase 4 (mm)	Wheelbase 5 (mm)
Motorcycle	1305–1695				
Urban/subcompact tourism	2425–2563				
Hatchback/sedan	2636–2770				
Big sedan	2820–2924				
Bus (two axles)	5770–6080				
Bus (three axles)	6090–7140	1350–1600			
Articulated bus (two axles)	5980–6028	6480–6540			
Industrial VAN	3000–4750				
Rigid truck (two axles)	4200–5950				
Rigid truck (three axles)	4200–4500	1250–1500			
Rigid truck (four axles)	1700–1900	2500–3000	1250–1500		
Articulated truck (four axles)	2990–3900	4250–6250	1200–1350		
Articulated truck (five axles)	2990–3900	4250–6250	1200–1350	1200–1350	
Articulated truck (six axles)	2990–3900	4250–6250	1200–1350	1200–1350	1200–1350

**Table 6 sensors-20-04597-t006:** Profile performance.

State	#Runs	Average (ms)	Shortest (ms)	Longest (ms)
Read Piezos	It depends on the speed of the vehicle, stop when piezo sensor 2 is driven over			
Analyze	10	7.9	7.4	8.3
Take photo	10	655	621	680

**Table 7 sensors-20-04597-t007:** Speed measurement error estimation.

Vehicle Speed (km/h)	Errordb (%)	Errorwave (%)	Errorfs (%)	Errorspeed (%)	Errorspeed (km/h)
25	0.013	0.012	0.012	0.024	0.006
50	0.013	0.023	0.023	0.036	0.018
75	0.013	0.035	0.035	0.048	0.036
100	0.013	0.046	0.046	0.059	0.059
125	0.013	0.058	0.058	0.071	0.088
150	0.013	0.069	0.069	0.082	0.123
175	0.013	0.081	0.081	0.094	0.164

**Table 8 sensors-20-04597-t008:** Wheelbase measurement error estimation.

Vehicle Speed (km/h)	Error_speed_ (%)	Error*_Delay_*_1,2_ (%)	Error_Wheelbase 1_ (%)	Error_Wheelbase 1_ (m)	Error*_Delay_*_2,3_ (%)	Error_Wheelbase 2_ (%)	Error_Wheelbase 2_ (m)
25	0.024	0.505	0.529	0.007	0.159	0.183	0.003
50	0.036	1.010	1.046	0.014	0.317	0.353	0.005
75	0.048	1.515	1.563	0.021	0476	0.524	0.007
100	0.059	2.020	2.079	0.029	0.635	0.694	0.010
125	0.071	2.525	2.596	0.036	0.794	0.864	0.012
150	0.082	3.030	3.113	0.043	0.952	1.035	0.014
175	0.094	3.535	3.629	0.050	1.111	1.205	0.017

**Table 9 sensors-20-04597-t009:** Classification results with real traffic.

	#Vehicles	Category “In Range”	Category “Unknown” Measurement OK	Category “Unknown” Measurement NO OK	Unknown (%)	Successful (%)
Test 1	42	38	0	4	10%	90%
Test 2	214	201	2	11	6%	94%
Test 3	61	60	0	1	2%	98%
Test 4	140	138	1	2	2%	98%

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
