# Peer review of "A Virtual Instrument for Road Vehicle Classification Based on Piezoelectric Transducers"

_sensors, 2020, doi:10.3390/s20164597_

Round 1

Reviewer 1 Report

In general, it is difficult to recommend this paper to be published in its current form. The authors should take into consideration the following points:

1 The authors should reorganize the paper in order to facilitate the reading. The motivation of the problem is not clear.

2. the author should do more to clarify the novelty content provided by the proposed method.

3. The Virtual Instrument is not given clearly.

Reviewer 2 Report

This paper studies the virtual instrument design problem for road vehicle classification based on Piezoelectric transducers, the paper has not been  well written in the opinion of this reviewer, the following are some points need to revised or clarified:

  1. What does the results given in FIgure 4 and Figure 7 mean?
  2. Are the hardwares presented in Section 3 designed by the authors? If not, I suggest it should be introduced briefly;
  3. Figure 11 is a little messy, it needs to be clarified further;
  4. In Section 4, the vehicle classification algorithm has not been proposed clearly, which is important.

Reviewer 3 Report

This paper presents a virtual instrument for road vehicle classification. The proposed method relies on several sensors, including magnetic sensor, piezo sensors, and camera. The experiments on the road with real traffic show that the proposed method can detect the road vehicles well. My concerns are as follows:

  • As an approach for in-vehicle systems, which intrinsically close to Automatic Driving Assistant Systems (ADAS), some recent state-of-the-art ADAS methods should be surveyed in this paper:

[r1] Automatic Dangerous Driving Intensity Analysis for Advanced Driver Assistance Systems From Multimodal Driving Signals, IEEE Sensors Journal

[r2] Multiframe-based high dynamic range monocular vision system for advanced driver assistance systems, IEEE Sensors Journal

[r3]UrbanMobilitySense: A user-centric participatory sensing system for transportation activity surveys, IEEE Sensors Journal

[r4] Driver Danger-Level Monitoring System Using Multi-Sourced Big Driving Data, IEEE Transactions on Intelligent Transportation System

[r5] Service-oriented dynamic connection management for software-defined Internet of vehicles, IEEE Transactions on Intelligent Transportation System

  • The vehicle category is determined by the number of axles and the wheelbase or distance between axles. This kind of strategy seems to be a little bit arbitrary. First, how do you deal with the deviation in one category? Second, how do you adjust the measuring error during the transmission between sensors?
  • A quantitative result of vehicle category classification is highly preferred. How is the accuracy and recall of the classification?
  • In order to employ such framework in real-life condition, it would be good to know about its processing/compute time. Please discuss time complexity of the proposed system.
  • This paper contains many ambiguous words and sentences. The grammar clearly needs improvement, and the paper should be proof read.

Round 2

Reviewer 1 Report

The authors have responded to my previous questions adequately, and can be accepted in the current version.

Reviewer 2 Report

This version of the paper is OK to me.

Reviewer 3 Report

The revised version well address all my concerns. I recommend acceptance.